# Nutrition in Cystic Fibrosis—Some Notes on the Fat Recommendations

**DOI:** 10.3390/nu14040853

**Published:** 2022-02-18

**Authors:** Birgitta Strandvik

**Affiliations:** Department of Biosciences and Nutrition, Karolinska Institutet, Huddinge, 14183 Stockholm, Sweden; birgitta.strandvik@ki.se; Tel.: +46-(0)70-5486348

**Keywords:** energy, arachidonic acid, DHA, linoleic acid, lipid mediator, CFTR modulators

## Abstract

Nutrition is important in cystic fibrosis (CF) because the disease is associated with a higher energy consumption, special nutritional deficiencies, and malabsorption mainly related to pancreatic insufficiency. The clinical course with deterioration of lung function has been shown to relate to nutrition. Despite general recommendation of high energy intake, the clinical deterioration is difficult to restrain suggesting that special needs have not been identified and specified. It is well-known that the CF phenotype is associated with lipid abnormalities, especially in the essential or conditionally essential fatty acids. This review will concentrate on the qualitative aspects of fat metabolism, which has mainly been neglected in dietary fat recommendations focusing on fat quantity. For more than 60 years it has been known and confirmed that the patients have a deficiency of linoleic acid, an n-6 essential fatty acid of importance for membrane structure and function. The ratio between arachidonic acid and docosahexaenoic acid, conditionally essential fatty acids of the n-6 and n-3 series, respectively, is often increased. The recently discovered relations between the CFTR modulators and lipid metabolism raise new interests in this field and together with new technology provide possibilities to specify further specify personalized therapy.

## 1. Introduction

Cystic fibrosis (CF), also known as mucoviscidosis, was identified in 1936–1938 as a special disease, cystic fibrosis of the pancreas, distinguished from celiac disease due to its characteristic pancreatic involvement and rapidly progressive pulmonary changes [1,2]. Before efficient pancreatic enzymes and a wide spectrum of antibiotics were available, most patients died in early childhood. The impressive progress of care for these patients during the latest decades [3,4], has now brought the adult to exceed the child patients in developed countries [5,6,7,8]), can be related to better clinical treatment, development of drugs and organ transplantation. The most common cause of death is the pulmonary destruction caused by the disease. The importance of nutrition for the pulmonary function is well documented [9,10,11].

Many papers have been published regarding nutritional recommendations for CF patients, as well as during recent years [12,13,14,15,16,17]. These documents unanimously recommend high energy, 110–200% of recommended daily intake (RDI), often with higher protein and fat contributions. This review will not further repeat these mostly concordant recommendations but discuss some special issues regarding the energy-rich macronutrient, the fat, whose quality has been given less attention in the guidelines as well as in evaluating the clinical results of the recommendations. Our knowledge about the well-known lipid abnormalities in CF will improve with the available new techniques giving possibilities to extensive lipidomics [18,19,20,21,22], expanding the knowledge about the role of lipids in the disease. In this context, the quality of fat intake is important since it is reflected in cell membranes influencing transport and metabolism [23,24,25].

## 2. Fat Recommendations and Intake

A difference in patient status between Boston and Toronto despite general similarities in treatment was reported by Corey et al. in 1988 [26]. The US, like many European countries, but differently from Toronto, recommended a low-fat diet due to the problems with steatorrhoea, probably at least partly related to less-efficient pancreatic enzyme restitution therapy (PERT). That paper resulted in a general recommendation of high-fat diet to patients with CF, and recent comparisons between the countries showed that the differences in patient status decreased [27] and were recently mainly omitted [28]. These results suggest that higher lipid intake is a stronger factor than only high energy intake.

A higher fat intake usually results in a high intake of saturated fat [29,30,31,32], which might be a factor explaining why many studies report that the high fat recommendations are not followed, sometimes only in half of the patient population [33,34]. It is difficult to obtain the high recommended energy intake without increasing the fat, it being the most energy dense food. In a study by Stallings et al. a structured lipid supplement with 54% polyunsaturated fat showed clinical improvement compared to a calorie- and macronu-trient-matched placebo group after 3 months treatment [35]. Of interest are a few studies showing that normal weight gain can be achieved in CF without increased energy intake if the essential fatty acid deficiency (EFAD) was treated [36,37]. So, although the high energy demand is multifactorial, the quality of the fat should command more interest than just the amount.

Patients with CF have a higher need (see below) for EFA, e.g., the fatty acids of the omega-6 (n-6) and omega-3 (n-3) series, which we cannot synthesize. A high fat intake usually also increases the intake of EFA [38,39]. That connection has generally not been commented on in the general recommendations but is of interest when comparing the intake of different fatty acids intake in different countries (Table 1). These intakes reflect also that of the children with the disease. It is well-known that the intake of n-6 fatty acids is high in the US [40] and that it also differs between European countries, as reflected in maternal plasma during pregnancy [41,42]. In this context it is of interest that the most recent dietary recommendations for CF from the US show no indications for even discussing recommendations of EFA [14]. To the contrary, it is usually emphasized that an increased intake of EFA cannot be recommended due to the shortcoming of good controlled randomized studies, which is true, but the low levels related to the severe CF phenotype cannot just be neglected [43]. Only a few recommendations comment on the possibility that EFA may be of value [17,44,45]. This is a little surprising since lipid abnormalities with deficiency of especially the n-6 fatty acid linoleic acid (LA, 18:2n-6) has been known for more than 60 years in patients with CF [46]. Importantly, any comments regarding EFA in the recommendations only refer to the n-3 fatty acids, although no recommendation is given because no convincing clinical effects are shown in many studies supplementing n-3 fatty acids [12,47,48,49,50].

## 3. Fatty Acid Abnormalities in CF

The old report about LA deficiency in CF has been confirmed in different blood compartments [51,52,53] and in various tissues [54,55,56,57]. Many studies of supplementation with LA have been performed, but were often of relatively short duration and, lacking in stable study conditions, not resulting in clear outcomes and motivating general recommendations. Some studies have shown an important influence on growth [37,58,59] and some observation studies have shown associations between lung function and serum phospholipid concentrations of LA [44,60,61,62,63]. In a longitudinal prospective observation study of neonatally screened patients, the Wisconsin group has shown a strong relation between the LA concentration and growth and pulmonary function up to 12 years of age [64,65,66]. Some other longitudinal studies have shown an influence on growth, renal and liver function [55,58,67,68]. The progress of CF-related liver disease has shown an association to low LA [69,70], and Sweden, where the largest centres had provided LA supplementation for decades [4], had the lowest percentage of portal hypertension and cirrhosis compared to 10 other countries worldwide [71].

LA is an important constituent of cell membranes and together with the ratio of n-6/n-3 fatty acids and cholesterol influences the function of transmembrane proteins, including ion channels and enzymes [23,72]. In the body, LA is transformed to arachidonic acid (20:4n-6, ARA) (Figure 1), which is the substrate for the pro-inflammatory eicosanoids, increased in CF [73,74], and important in the defense of infections but also involved in many metabolic processes. The synthesis of ARA is the rate-limiting step in eicosanoid production [75]. There seems to be a feedback mechanism since the relation between ARA and LA is inverted [20], signaling possible importance in the context of prostaglandin production. This inverse relation was observed in cell systems and related to an inhibition of FADS2 by LA [76]. That is supported by studies of rats showing that EFA deficiency increases FADS 2 activity [77]. The enzyme activities involved in the transformation of LA are increased in CF, suggesting it to be more basically related to CFTR [78] than being an effect of the increased AA release [79]. Since the AA release in CF blood cells could not be inhibited by dexamethasone [80] and not by bradykinin in cell systems [81,82], it is an open question if the high turnover is an effect of, or a cause for, the highly increased AA release [83,84,85] on the one hand, or on the other if the LA deficiency itself is increasing the turnover [77]. Algorithms have been presented to suggest relations between the disturbed lipid metabolism and the clinical symptoms, which later clinical investigations have partly supported [86,87].

The most important EFA of the n-3 fatty acid series is docosahexaenoic acid (22:6n-3, DHA). DHA is often decreased in CF, and low concentrations are related to the liver disease [88,89]. It is of interest that in CF mice receiving long-term treatment with high doses of DHA, liver disease could be attenuated [90]. The improvement of ileal mucosa and pancreas pathology in CF mice by high doses of DHA has not been confirmed [91] but has increased the interest for studies of supplying DHA to patients but without resulting in clear clinical improvements [92]. This long-chain n-3 PUFA also contributes to membrane function by interfering with raft formation important for transport function in membranes [93]. DHA, like other long-chain n-3 fatty acids, are alo transferred to special pro-solving lipid mediators (SPM), e.g., the resolvins, maresins and protectins, of importance to balance inflammation provided by the ARA metabolites [94]. The role of SPM in regulating autophagy, which is defective in CF, may be a possible therapeutic option against 136 the persistent inflammation [95,96,97]. That was supported by adding resolvin D1 to infected CF KO mice and in studies of cells from CF patients [98]. Since long-term administration of DHA did not improve clinical status [99], it has been suggested that impaired production of SPM in CF is associated with abnormal expression of lipoxygenases [95]. Furthermore, lipoxin A4, an anti-inflammatory SPM from ARA, is deficient in CF [100]. Altogether, the lipid metabolism of polyunsaturated fatty acids is unbalanced in CF [20,51].

Reasons for the low interest in these well-known fatty acid abnormalities in CF might be multiplex: (1) the unclear connection to the genetic defect, the defective chloride and bicarbonate channel CFTR (Cystic fibrosis transmembrane conductance regulator) [101] compounded by the fact the lipid abnormalities are related to a more severe phenotype [43,102]. Today, more than 2000 mutations are identified, but only a minority are related to the CF phenotype, although classified in six types, some with very minor clinical relevance [103,104]. New technologies show different lipid patterns related to type of mutations [105], and correlations between different phospholipid abnormalities and mild and severe clinical disease [106]. Since severe disease is related to mutations with more bacterial infections some of these changes can be due to bacterial interactions, which needs further studies for differentiation. (2) The levels or concentrations in the serum/plasma of LA are often low but seldom at a real deficient status, and tissue levels/concentrations are usually only obtained at autopsy [55,57], but can be shown also in nasal and intestinal biopsies [54]. Studies with contemporary examinations of plasma and tissues show that tissues can have varying and much lower concentrations than in plasma [55,107], suggesting that relatively low blood concentrations might also be of clinical relevance. Lower concentrations and values in different tissues would also explain why the EFA index (the ratio between Mead acid (20:3n-9) to ARA, i.e., the triene/tetraene (T:T) ratio) may remain pathological, when plasma levels are normalized. This is reported from Belgium [108,109], which has one of the highest fat intakes in Europe (Table 1). (3) The relation to other fatty acid abnormalities is partly unclear, e.g., the high ratio between ARA and DHA, although not always related to low DHA or high ARA, is not fully explained. Nor is the probably compensatory increase in the monounsaturated fatty acids palmitoleic acid (16:1n-7), oleic acid (18:1n-9) and Mead acid [53,78,110,111]. (4) The relation to other lipid abnormalities such as low cholesterol and different pattern of the lipoproteins [112,113,114] are incompletely understood. (5) The difficulty to identify clinically the deficiency in relation to symptoms, such as growth failure, pancreatic insufficiency increased resting respiratory expenditure, etc. and not least. (6) the omission to check the concentration in blood, which recently was shown only to be performed only in 15% of centers caring for CF patients according to the DIGEST (Developing Innovative Gastro Enterology Special Training) program [115].

The increased activity of the epithelial sodium channel (ENaC), which is considered to contribute to the pulmonary symptoms, have been suggested to be influenced by LA [116]. ENaC are expressed in the kidney, and the renal excretion of sodium was restored after long-term LA supplementation [67]. Sodium excretion in sweat was also decreased after long-term LA supplementation [59,74,117]. Other transporters and membrane-bound activities might also be influenced, since the Na(+)–K+ ratio in erythrocytes was normalized by regular LA supplementation [118].

## 4. Is Dietary Qualitative Intervention Justified?

Altogether, few dietary recommendations in CF mention the well-known deficiencies of EFA and the different needs they satisfy, e.g., the n-6 for the possible membrane physiology and its products in balancing inflammation and infections and the n-3 also involved in membrane physiology and for anti-inflammatory balancing function via SPM [92]. At least three studies use n-6-rich oil as control when supplying n-3 fatty acids and see no differences between the groups [119,120,121]. In a recent systematic review of supplementations in CF, LA supplementation was summarized to influence nutritional status and n-3 supplementation to reduce inflammatory markers [122], supporting the theoretical concept that both n-6 and n-3 fatty acids might be valuable in CF nutrition.

Despite extensive research during more than 30 years since CFTR was detected, no cure of the disease has been developed. In the search for therapy, the high-throughput technique had identified correctors in the synthesis of some mutations and especially one potentiator, ivacaftor, for the action of some mutated proteins making it possible to improve the status of several patients [123]. Combinations of the potentiator and correctors have further increased the potential target of mutations and shown to influence the lung function, sweat test and weight gain but not less the chronic inflammation [124,125]. Of certain interest is that the modulators are recommended to be ingested together with lipids [126]. The real influence of CFTR mutations on clinical pathophysiology is still inconclusive, and many other types of modulators are under investigation [127], which illustrates the persistent challenges of clinical treatment. It is also important to realize that a general clinical improvement in lung function with modern treatment was obvious before the recent development of modulators [128].

To ignore the deficiency of fatty acids, as being of well-documented importance for cell physiology, is hopefully coming to an end since recent research has shown a strong difference in the lipid pattern related to the type of mutation [105]. Furthermore, it is now shown that the modulators interfere with the membrane lipid structure, concerning the glycerophospholipids, the sphingomyelins, and also the ceramides [18,19,21]. All repeatedly shown involvement in the CF context [106,129,130]. An influence on the prostanoids has also been suggested [131]. These findings open a link between the CFTR and the lipid abnormality, which has been lacking and probably contributed to the neglect of treating the EFAD in CF. A further reason for considering the EFA abnormalities is that there might be rather extensive deficiency in different organs since serum concentrations of the fatty acids are less reflecting the total body metabolism, and it is therefore a need for longer treatment periods to influence organ symptoms [67,68].

An ongoing multicenter European study blindly supplying either LA or oleic acid for one year might hopefully contribute to an answer on the putative beneficial effect of LA supplementation in CF (NCT 04531410) and, by using metabolomics, further add to the knowledge of the pathophysiology [132]. More studies are warranted.

## 5. Conclusions

The general guidelines worldwide recommend high energy intake with special attention to protein and fat, but the fat recommendations are seldom achieved. The type of fat should be given more attention in relation to the known deficiencies and the conditions of the patients. More normal energy recommendations can probably be obtained if the quality of the fat supply is changed with greater focus on linoleic-rich oils, such as sunflower oil and corn oil, as opposed to the general recommendation of canola oil and olive oil for the normal population. Fish or omega-3 supplementation should also be encouraged potentially to hamper the inflammation. Such recommendations would be valuable non-toxic additives to the modulator therapy in waiting for long-term controlled studies.

## Figures and Tables

**Figure 1 nutrients-14-00853-f001:**
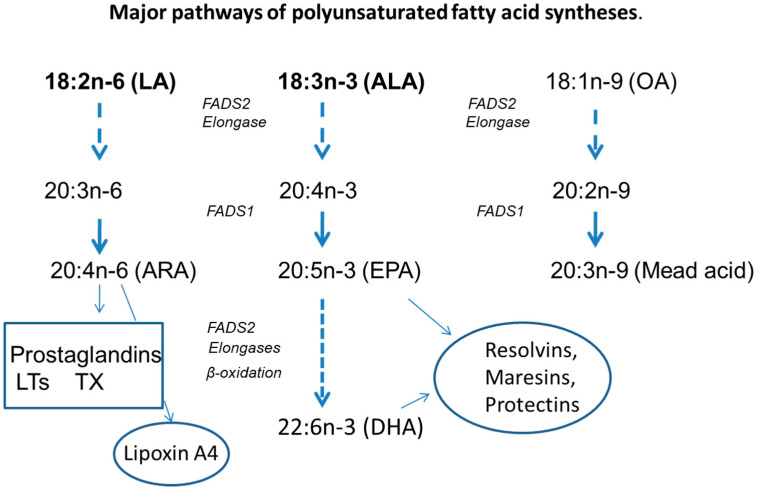
Major pathways of polyunsaturated fatty acid synthesis. Simplified major transformation steps to long-chain polyunsaturated fatty acids from the essential fatty acids, linoleic (LA) and α-linolenic (ALA) acids and the endogenously synthesized oleic acid (OA), which all compete for the same desaturases (FADS1 and FADS2) and elongases. The major metabolites are indicated in boxes, proinflammatory and ellipses (anti-inflammatory) lipid mediators. ARA, arachidonic acid; EPA, eicosapentaenoic acid; DHA, docosahexaenoic acid; LT, leukotrienes; TX, tromboxanes.

**Table 1 nutrients-14-00853-t001:** Calculated fat intake in g/capita/day of commonly used vegetable oils, dairy fats, raw animal fat and fish (sum of demersal, pelagic and other marine fishes) in the USA, Canada, the United Kingdom, Belgium, The Netherlands, Spain, Italy and Sweden according to the Food and Agriculture Organization of the United Nations (FAO) 2019 (www.fao.org/faostat/en/#data/FBS, accessed on: 9 February 2022). + indicates that information aout butter is missing.

Fat	USA	Canada	UK	Belgium	Netherlands	Italy	Spain	Sweden
Soybean oil	61.8	1.79	13.5	11.6	11.6	10.6	14.1	0.16
Sunflower oil	0.52	0.94	7.07	9.54	1.69	31.6	11.7	6.99
Corn oil	6.67	2.79	1.25	4.82	1.83	1.61	2.88	0.16
Olive oil	2.67	3.00	2.67	3.42	2.30	30.9	27.2	2.48
Ratio n-6 richoils/olive oil	25.8	1.84	8.17	7.60	6.73	1.42	1.05	2.95
Cream + milk	24.2	19.6	21.2	33.6	33.2	17.0	19.2	28.6
Butter	5.41	8.27	6.43	ND	13.31	5.5	2.78	11.3
Animal fat, raw	5.84	12.9	5.66	26.7	10.8	2.47	8.45	5.87
Fish	1.17	1.12	2.29	1.85	1.51	2.94	2.14	1.50
Ratio n-6 rich oils/ dairy + animal fat	2.30	0.17	0.81	0.43	0.35	2.25	1.04	0.21
Ratio n-6 rich oils/ dairy + butter + animal fat	1.95	0.14	0.66	ND	0.27	1.75	0.94	0.16
Total fat, (based on presented fat categories)	106	47.4	57.4	88.1+	74.3	71.7	61.3	54.6

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
