# Peer review of "Nutrition in Cystic Fibrosis—Some Notes on the Fat Recommendations"

_nutrients, 2022, doi:10.3390/nu14040853_

Round 1

Reviewer 1 Report

Review report

  • A brief summary 

The present commentary titled “Nutrition in Cystic Fibrosis – Some Notes on the Fat Recommendationsis focused on the qualitative aspects of lipid metabolism in cystic fibrosis and highlights the importance of nutrition. The new interest in this field is based on the possibilities for specified personalized therapy.

  • General concept comments

Nevertheless that reviews for nutrition in fibrous necrosis exist in the literature, the present review emphasizes the fatty acid abnormalities in CF and update the information about the importance of the specific fatty acids for cell membranes and for the body defense system and complication of CF as liver disease, pulmonary symptoms, and their (FA) concentration in blood could be used as critical markers for severity of CF. The authors underline the importance of lipid nutrition in CF as a critical factor for influencing both lung function, weight gain, even chronic inflammation. The weakness of the review is that the percentage of new cited references is very low (within the last 5 years). The authors should reconsider including more recent references.

  • Specific comments 

The text in Figure 1 should be of higher quality.

Author Response

Thank you for kind review. There were 37% of the references within the latest 5 years, which might look small amount but old research is not always of less quality than new ones, and there is one explanation.

The fat abnormalities were ranked uninteresting when the gene was detected 1989, and all research focused on proteomics in the enthusiasm that the riddle of CF would be solved within some years. Now when reality shows that no cure has been found in more than 30 years and the fantastic results of high-throughput technology have identified modulators which influence the disease, although unspecific by modulating lipids, new interest has developed in the old knowledge of disturbed lipid metabolism. Therefore it is impossible to make the story clear without referring to old research from time before the gene discovery. Most recent references are also from the latest 2-3 years which illustrates this reality. I have now added further 10 references from these recent years and hope that that will be satisfactory.

The text in Table 1 has been edited but was changed by the editors before and therefore I could only make smaller changes.

Reviewer 2 Report

Strandvik comments on lipid abnormalities, an important and often overlooked detail of cystic fibrosis critical care. This review focuses on qualitative aspects of fat metabolism, whereas most published information regarding lipid abnormalities in CF is concerned with fat quantity. The review is of particular importance as the recently developed CF modulators have been shown to affect lipid metabolism. Strandvik's commentary could provide guidelines to clinicians when recommending dietary plans to CF patients.

The commentary is thoughtfully written and researched. My only concern is that the use of the English language should be improved throughout. Although major editing is not needed, the commentary would read much easier if revised by an editor.

Author Response

Thank you for your review and kind remarks.

The English has now been revised by prof Hugh Beach at Uppsala University, Sweden. I hope that is satisfactory (in red in the revised manuscript). The blue changes are related to the incorporation of further references.